# Transcriptomic Profiling of Buds Unveils Insights into Floral Initiation in Tea-Oil Tree (*Camellia oleifera* ‘changlin53’)

**DOI:** 10.3390/plants14152348

**Published:** 2025-07-30

**Authors:** Hongyan Guo, Zongshun Zhou, Jian Zhou, Chao Yan, Wenbin Zhong, Chang Li, Ying Jiang, Yaqi Yuan, Linqing Cao, Wenting Pan, Jinfeng Wang, Jia Wang, Tieding He, Yikai Hua, Yisi Liu, Lixian Cao, Chuansong Chen

**Affiliations:** 1Key Laboratory of Cultivation and Utilization for Oil-Camellia Resources, Experimental Center for Subtropical Forestry, Chinese Academy of Forestry, Xinyu 336600, China; guohongyan5@caf.ac.cn (H.G.); zhouzongshun24@163.com (Z.Z.); yanc01@163.com (C.Y.); zwb7234@163.com (W.Z.); 18279323430@163.com (Y.J.); yyq1116@163.com (Y.Y.); caolq1991@126.com (L.C.); pan_ada@126.com (W.P.); 15084771361@163.com (J.W.); wangjia6025@126.com (J.W.); heytieding@163.com (T.H.); 18907902533@163.com (Y.H.); 13979080267@163.com (Y.L.); caolixian0914@163.com (L.C.); 2Xinyu University, Xinyu 336600, China; zhj20231222@163.com; 3Jiangxi Environmental Engineering Vocational College, Ganzhou 341000, China; lichang00052@163.com

**Keywords:** *Camellia oleifera*, floral initiation, buds, transcriptome analysis, gibberellins

## Abstract

Flowering is a key agronomic trait that directly influences the yield of the tea-oil tree (*Camellia oleifera*). Floral initiation, which precedes flower bud differentiation, represents a critical developmental stage affecting the flowering outcomes. However, the molecular mechanisms underlying floral initiation in *C. oleifera* remain poorly understood. In this study, buds from five key developmental stages of a 12-year-old *C. oleifera* cultivar ‘changlin53’ were collected as experimental samples. Scanning electron microscopy was employed to identify the stage of floral initiation. UPLC-MS/MS was used to analyze endogenous gibberellin (GA) concentrations, while transcriptomic analysis was performed to reveal the underlying transcriptional regulatory network. Six GA types were detected during floral initiation and petal development. GA_4_ was exclusively detected at the sprouting stage (BII), while GA_3_ was present in all samples but was significantly lower in BII and the flower bud primordium formation stage (BIII) than in the other samples. A total of 64 differentially expressed genes were concurrently enriched in flower development, reproductive shoot system development, and shoot system development. Weighted gene co-expression network analysis (WGCNA) identified eight specific modules significantly associated with different developmental stages. The magenta module, containing Unigene0084708 (*CoFT*) and Unigene0037067 (*CoLEAFY*), emerged as a key regulatory module driving floral initiation. Additionally, *GA20OX1* and *GA2OX8* were identified as candidate genes involved in GA-mediated regulation of floral initiation. Based on morphological and transcriptomic analyses, we conclude that floral initiation of *C. oleifera* is a continuous regulatory process governed by multiple genes, with the *FT*-*LFY* module playing a central role in the transition from apical meristem to floral meristem.

## 1. Introduction

*Camellia oleifera* Abel. (*C. oleifera*) is an evergreen tree species cultivated in subtropical regions and is one among the four major woody oil plants globally, along with oil palm, olive, and coconut [1,2]. Because of geographical and climatic constraints, *C. oleifera* is widely distributed in the subtropical mountainous regions of the Yangtze River Basin [3]. Its seeds yield tea oil rich in oleic acid and natural antioxidants, with notable nutritional and health benefits [4,5,6]. Flowering is a critical agronomic trait affecting *C. oleifera* yield, and floral initiation plays a pivotal role in determining flowering time [7].

Floral initiation is influenced by both external and internal signals. Environmental factors, such as photoperiod, temperature, nutrient status, drought, salinity, exogenous hormones and chemicals, and microbial pathogens, as well as endogenous signals, such as plant age, hormone levels, and carbohydrate status, all contribute to its regulation [8,9,10,11,12,13]. These cues are perceived primarily in the leaves and shoot apical meristem [13,14]. In 1-year-old herbaceous plants, floral initiation marks the transition from vegetative to reproductive growth. Molecular studies on this process have largely focused on model herbs such as *Arabidopsis thaliana* and rice. In *A. thaliana*, flowering is regulated by a complex network involving more than 100 genes across multiple pathways: photoperiod, ambient temperature, autonomous, integrator, gibberellins (GAs), and vernalization [15,16]. By contrast, floral initiation in woody perennials occurs in two distinct phases: the initial transition from vegetative to reproductive growth in the life cycle (marking maturity and the capacity to flower) and the annual initiation of floral structures post-maturity. Unlike annuals—where flowering is terminal and leads to senescence—perennials undergo repeated cycles of vegetative and reproductive growth. Hence, annuals and perennials have different flowering traits.

In Arabidopsis, the *FT* gene promotes flowering and is upregulated by *CONSTANS* (*CO*), which is sensitive to photoperiods [17,18]. *FT* is repressed by *FLOWERING LOCUS C (FLC)*, which delays flowering by blocking gene transcription in photoperiodic flowering. The attenuation of *FLC*, a key floral repressor, distinguishes summer- and winter-annual phenotypes [19,20]. In perennial poplar, functionally differentiated *FT1* and *FT2* separate the timing of reproductive initiation and vegetative growth to different seasons. *FT1* rises in response to low temperatures of winter and promotes reproduction, and *FT2* is induced by exposure to long days and warm temperatures of spring and early summer and promotes vegetative growth [21]. *LEAFY* (*LFY*), a primary activator of floral meristem development, plays a crucial role in determining floral transition timing. Its expression is promoted by *SQUAMOSA PROMOTER BINDING-LIKE* (*SPL*) transcription factors (e.g., *SPL9*) and GA-sensitive DELLA proteins [22,23,24]. Interestingly, *LFY* expression is high in alternate-bearing mango (*Mangifera indica* L.) cultivars during flowering but inhibited in ever-flowering longan cultivar ‘Sijimi’ [25,26]. These examples underscore the mechanistic differences in floral initiation between woody and herbaceous plants. Due to long generation times and complex genetic backgrounds, the molecular mechanisms of flowering in woody perennials remain less explored than in model herbs.

In *C. oleifera*, flowering is essential for yield, yet its regulation is complex. Flower buds typically form at the base of vegetative buds located at the terminals of spring shoots or in the axils of new leaves [27]. These spring shoots originate from vegetative buds formed in the previous year, which themselves are located at the terminals or leaf axils of older shoots. Flower bud differentiation in *C. oleifera* has been reported to occur between early April and late July, with flowering taking place from early October to the end of December. The timing of bud differentiation and flowering varies among cultivars. Floral initiation precedes flower bud differentiation and is a critical developmental stage influencing *C. oleifera* flowering. We revealed that old leaves in *C. oleifera* serve as key photoperiodically sensitive organs, promoting *FT* expression through light-responsive mechanisms to initiate floral bud formation. The transduction of endogenous signaling molecules from old leaves to axillary buds is required to initiate floral transition through induced transcriptional activation of floral meristem-specific genes. However, the specific genetic determinants mediating this developmental shift remain to be systematically characterized through comprehensive transcriptomic analysis, and the following question remains: which genes, *LFY*, APETALA1 (*AP1*) or any other genes, in the buds were activated? 

## 2. Results

### 2.1. Morphological Characteristics of Floral Initiation in C. oleifera at Different Developmental Stages

The formation of flower bud primordia indicates that the floral meristem has been activated, and floral initiation has begun [28]. Buds are key organs during floral initiation; therefore, buds at five key developmental stages were selected for transcriptome analysis: dormancy stage (stage I), sprouting stage (stage II), flower bud primordium formation stage (stage III), petal differentiation stage (stage IV), and completion of petal differentiation stage (stage V) (Figure 1).

### 2.2. Quantitative Analysis of Endogenous Gibberellins in Buds at Different Developmental Stages

To investigate the involvement of GAs in floral initiation, we quantified endogenous GA levels in *C. oleifera* buds across five developmental stages. Of the 10 GA types assessed (GA_1_, GA_3_, GA_4_, GA_7_, GA_9_, GA_15_, GA_19_, GA_20_, GA_24_, and GA_53_), only 6 GAs—GA_3_, GA_4_, GA_9_, GA_15_, GA_19_, and GA_20_—were detected (Figure 2a). The total GA content was significantly higher in stages BII, BIII, and BV than in stages BI and BIV (Figure 2a). The dominant GA types in each stage were as follows: GA_3_ in BI and BV, GA_19_ in BIV, and GA_20_ in BII and BIII (Figure 2b). GA_3_ and GA_4_ are the biologically active forms of GA gibberellin. Notably, bioactive GA (GA_3_ + GA_4_) levels in BI, BII, BIII, and BIV were significantly lower than those in BV. GA_4_ was exclusively detected in BII, while GA_3_ was present in all stages, albeit at significantly lower levels in BII and BIII. These findings suggest that floral initiation in *C. oleifera* requires only a relatively low concentration of active GAs, whereas a higher concentration is essential for petal differentiation.

### 2.3. Transcriptome Assembly and Annotation

Transcriptomic analysis using RNA sequencing (RNA-seq) was performed to investigate the gene regulatory networks underlying floral initiation across the five developmental stages of *C. oleifera* buds. An overview of the RNA-seq samples and clean read counts is shown in Table 1. The proportion of mapped reads per library ranged from 79.05% to 83.41% (Table 1). Sequencing quality metrics were high, with Q30 values (sequences with a sequencing error rate of <0.1%) of >91% and Q20 values (sequences with a sequencing error rate of <0.1%) of >97%. The mean GC content across all samples ranged from 45.88% to 47.56%, indicating satisfactory transcriptome assembly quality.

Principal component analysis revealed that the 15 samples could be grouped into 3 clusters: BI, BII, and BIII–V (Figure 3a). Samples from stages BIII, BIV, and BV were clustered together, suggesting similar transcriptomic profiles. Transcript annotation was conducted using BLASTx (version 2.2.2.1) against several protein databases—NCBI non-redundant (Nr), Swiss-Prot, Clusters of Orthologous Groups, Gene Ontology (GO), and Kyoto Encyclopedia of Genes and Genomes (KEGG)—with an E-value threshold of 10^−5^. A total of 27468 unigenes were annotated across all databases, while 60632 unigenes (79.7%) were annotated in at least one database.

### 2.4. Differentially Expressed Genes at Different Developmental Stages

To identify genes associated with floral initiation, 10 pairwise comparisons of gene expression between stages were conducted. Differentially expressed genes (DEGs) were defined as those with |log2 (Fold Change)| > 1, a false discovery rate (FDR) < 0.05, and RPKM < 2 in all samples. A total of 26996 DEGs were identified across all comparisons. Notably, comparisons of BII vs. BI, BIII vs. BI, BIV vs. BI, and BV vs. BI each revealed more than 14,000 DEGs (Figure 2b), indicating substantial transcriptional differences between the dormancy stage (BI) and later stages (Figure 3b). Conversely, only 292 DEGs (173 upregulated and 119 downregulated) were identified in the BV vs. BIV comparison (Figure 3b), suggesting minimal transcription differences between these two stages. This indicates that stages IV and V share similar developmental characteristics. A comparison of common and unique DEGs across the 10 comparisons is shown in Figure 3c. Only three genes were differentially expressed in all comparisons. The absence of unique DEGs in the BV vs. BIV comparison supports the conclusion that transcriptional activity is highly conserved between these final stages.

### 2.5. GO Enrichment Analysis of DEGs

To gain further insight into the DEGs, GO enrichment analysis was performed. Across all comparisons, 2457 GO terms were identified, including 50 Level-2 GO terms and 3 Level-1 GO terms distributed among the three major GO categories: biological process (BP), cellular component (CC), and molecular function (MF) (Figure 4a). Among the top 20 significantly enriched terms, 5 were classified as BP, 6 as CC, and 9 as MF (Figure 4b). As BPs are more directly related to floral initiation than CCs and MFs, these were analyzed in greater depth. A total of 6163 DEGs were enriched in 1606 BP GO terms, with the top 21 shown in Figure 4c. These primarily included biological regulation (GO:0065007), phenylpropanoid metabolic process (GO:0009698), developmental process (GO:0032502), flower development (GO:0009908), reproductive shoot system development (GO:0090567), and shoot system development (GO:0048367). Because floral initiation in *C. oleifera* occurs in the shoot apex or leaf axils, DEGs enriched in flower development, reproductive shoot system development, and shoot system development were further examined. The analysis revealed 65, 84, and 106 DEGs enriched in these respective categories. A Venn diagram showed that 64 DEGs were commonly enriched across all three processes, suggesting that floral initiation, flower development, and spring shoot growth are co-regulated by the same set of genes (Appendix A). This supports the hypothesis that floral initiation and shoot development may occur simultaneously in *C. oleifera* (Appendix A).

### 2.6. KEGG Enrichment Analysis of DEGs

To further understand the significant DEGs identified, a KEGG pathway enrichment analysis was conducted. DEGs from 10 pairwise comparisons were categorized into 135 KEGG pathways, which were further grouped into 19 Level-B and 5 Level-A KEGG pathways (Figure 5a). The highest number of DEGs was associated with “Global and Overview Maps,” followed by pathways involved in carbohydrate metabolism and translation. Among the 135 KEGG pathways, 30 were significantly enriched (*p* < 0.05), and 18 of those were highly significantly enriched (*p* < 0.01). Analysis of the top 20 significantly enriched KEGG pathways revealed major involvement in plant hormone signal transduction, biosynthesis of secondary metabolites, flavonoid biosynthesis, phenylpropanoid biosynthesis, metabolic pathways, circadian rhythm plants, and starch and sucrose metabolism (Figure 5b). Specifically, 50 DEGs were related to the circadian rhythm plant pathway (Appendix A), and 261 DEGs were associated with plant hormone signal transduction (Appendix A). These findings suggest that circadian rhythm and hormone signaling likely play important roles in the floral initiation process of *C. oleifera*.

### 2.7. WGCNA Analysis of DEGs

To gain a comprehensive understanding of gene expression throughout the developmental stages of floral initiation and petal differentiation—and to identify key genes associated with floral initiation—a WGCNA was performed. From the 26,996 DEGs identified across the 10 comparisons, genes with low variation (MDA < 0.1) or excessive missing data (>10% samples) were filtered out. Ultimately, 25,267 genes with non-zero variance were retained. A co-expression network was constructed based on pairwise correlations of gene expression across samples. Highly correlated genes were grouped into modules, with smaller similar modules merged using a module merging threshold of 0.15 (Figure 6a). A total of 18 modules were generated, and the number of DEGs in each specific module is shown in Figure 6b.

### 2.8. Identification of Core DEGs at Each Developmental Stage

To identify modules specifically associated with each developmental stage, a labeled heatmap was used to analyze the correlation between modules and sample stages. Modules with correlation coefficients above 0.90 and *p* < 0.01 were considered stage-specific, resulting in the identification of eight specific modules. Notably, the turquoise, brown, green, grey60, and midnightblue modules showed significant positive correlations with BI (dormancy), BII (sprouting), BIII (flower bud primordium formation), BIV (petal differentiation), and BV (completion of petal differentiation), respectively. These are marked with red underlines in Figure 7a. Conversely, the black and lightcyan modules were significantly negatively correlated with BI, and the purple module was negatively correlated with BII, indicated with blue underlines. The DEG information of the top seven K_ME_ in these specific modules is shown in Appendix A.To explore the functional roles of these modules, GO enrichment analysis was performed on the DEGs within each. In the turquoise module, 1180 GO BP terms were enriched, with 121 GO terms being statistically significant (*p* < 0.05), including 9 terms related to flowering (*p* < 0.05). A total of 75 candidate DGEs enriched in these flowering-related terms were identified. Based on the WGCNA network relationships, a co-expression network was built using Cytoscape, revealing two core candidate genes (Appendix A). Applying the same method to other modules, five and three core candidate genes were identified in the brown and green modules, respectively (Appendix A). However, no core floral initiation-related DEGs were found in the grey60 or midnightblue modules.

### 2.9. Identification of Core DEGs Related to Floral Initiation During Whole Floral Initiation

Because floral development is a continuous process, we also focused on DEGs with consistent expression trends during floral initiation. According to the WGCNA analysis, the DEGs in the magenta module stood out. This module contained 722 DEGs enriched into 510 GO BP terms, of which 99 were significantly enriched. Nine of these GO terms were directly related to floral initiation: flower development (GO:0009908), reproductive shoot system development (GO:0090567), reproductive structure development (GO:0048608), reproductive system development (GO:0061458), negative regulation of flower development (GO:0009910), negative regulation of the reproductive process (GO:2000242), floral organ development (GO:0048437), floral organ formation (GO:0048449), and floral organ morphogenesis (GO:0048444). A total of eight candidate DEGs significantly enriched in these nine GO terms were identified (Appendix A). A co-expression network constructed in Cytoscape based on WGCNA relationship pairs revealed two key candidate genes: *Unigene0084708* (*CoFT*) and *Unigene0037067* (*CoLEAFY*) (Figure 7b). Meanwhile, the upstream regulatory gene *Unigene0057909* (*CoCO-like*) and two interacting genes of *CoFT—Unigene0007119* (*CoFD*) and *Unigene0031183* (*CoFD*)—were also present in the magenta module (Figure 7c). Together, these findings suggest that the CoCO-like-CoFT signaling pathway plays a crucial regulatory role during the floral initiation process in *C. oleifera*.

### 2.10. Identification of the DEGs Involved in Gibberellin Biosynthesis and Signal Transduction

GAs, a class of phytohormones, play a crucial role in floral initiation. KEGG enrichment analysis of DEGs revealed significant enrichment in plant hormone signaling pathways. Therefore, we further analyzed DEGs associated with GA biosynthesis and signal transduction. In the GA biosynthesis pathway, we identified eight DEGs involved in regulating the synthesis of various GAs, including ent-copalyl diphosphate; ent-kaurene; ent-7alpha-hydroxykaur-16-en-19-oic acid; and GA_15_, GA_9_, GA_19_, GA_20_, GA_3_, and GA_4_. In the GA signaling pathway, 12 DEGs were associated with GA signal transduction, namely, 3 *GID1*, 3 *DELLA*, 1 *GID2*, and 5 *B-ARR* genes.

### 2.11. Quantitative Real-Time PCR Analysis

Quantitative real-time PCR (qRT-PCR) analysis was conducted to validate the accuracy of the RNA-seq data. Twelve floral initiation-related DEGs were selected for qRT-PCR analysis across all samples. The expression patterns identified through qRT-PCR were consistent with those obtained from RNA-seq (Appendix A), confirming the reliability of the transcriptomic data.

## 3. Discussion

### 3.1. Reproductive Shoot Development Is a Key Component of the Reproductive Growth Initiation in the Annual Growth Cycle of Adult C. oleifera

Flowering is a pivotal phase in plant reproductive development, determining seed production and species propagation [27]. Flower formation begins when the vegetative stem apical meristem transitions to a reproductive meristem, which differentiates into a floral meristem and eventually gives rise to floral organs under the coordinated regulation of development signals and floral organ identity genes [14,29]. The regulation mechanisms of flowering differ between annual herbaceous and perennial woody plants [7]. In some woody perennials such as poplar, apple, citrus, and pear [21,30,31,32,33,34], flower bud differentiation occurs 1 year before blooming, with buds entering a dormancy phase before flowering the following year. By contrast, other species such as longan and mango initiate and complete flower development within the same year or even flower multiple times annually [25,35,36]. Thus, there are differences in the regulation of flowering traits between woody plants. In *C. oleifera*, floral bud primordia form at terminal or axillary buds during the growth of spring shoots and leaf expansion [28]. These spring shoots, serving as the primary flowering and fruiting structures, are identified as reproductive shoots. In the present study, 65 DEGs were implicated in both flower development and reproductive shoot development. Transcriptomic analysis confirmed that floral initiation and spring shoot growth in *C. oleifera* occur concurrently. Thus, reproductive shoot development is an integral component of reproductive growth initiation during the annual growth cycle. Based on morphological and transcriptomic findings, we propose that floral initiation and reproductive shoot growth originate simultaneously and spatially coincide.

### 3.2. The Photoperiod Pathway Involving CoCO-like, CoFT, and CoLFY Promotes Stem Apex Meristem Differentiation and Floral Initiation in C. oleifera

Extensive research has established that floral transition in plants is regulated by complex interactions between internal genetic factors and external environmental stimuli. Among these, the *CO*-*FT* signaling axis plays a central regulatory role [13,37]. Meanwhile, *LFY* acts as a master regulator of floral identity and is transcriptionally regulated by both *TFL1* (a repressor) and FT (an activator) [32,38]. *FT* competes with *TFL1* to form a complex with *TFL1*, thereby activating *LFY* expression and promoting the floral transition [39]. In *C. oleifera*, the regulatory role of *FT* in floral induction has been demonstrated experimentally [40,41,42]. Our results revealed similar expression patterns of *CoCO-like*, *CoFD*, *CoFT*, and *CoLFY*, suggesting that this photoperiod-regulated module orchestrates floral induction by integrating exogenous environmental cues into endogenous developmental programs. This *CO*-*FT*-*LFY* module likely plays a conserved and central role in floral regulation in *C. oleifera*.

### 3.3. GA_4_ and GA_3_ Might Play Different Roles in the Floral Initiation of C. oleifera

GA signaling is one of the principal floral pathways regulating flowering in plants [43]. Although many early studies have revealed the large chemodiversity of GAs in plants, only GA_1_, GA_3_, GA_4_, and GA_7_ are biologically active, controlling plant development [44,45,46]. The GA signaling pathway involves four key components, bioactive GAs, the GA-receptor GA INSENSITIVE DWARF1 (GID1), the central repressor DELLA, and the specific F-box protein SLEEPY1 (SLY1), which facilitates DELLA degradation [47]. This pathway is closely associated with endogenous GA levels in plants [45,46,48]. In chrysanthemums, elevated endogenous GA levels accelerate floral transitions from the juvenile to adult stage and promote early flowering [49]. In peach, bud sensitivity to GAs varies across developmental stages [50], and in tree peony, GA_3_ more effectively promotes endodormancy release than GA_4_ [51]. Thus, both GA type and concentration, along with development timing, are critical in GA-mediated floral regulation. In our work, six GAs were detected during floral initiation and petal development, but only GA_3_ and GA_4_ were bioactive. Noteworthy, GA_4_ was exclusively detected during the sprouting stage, along with its precursor GA_9_, suggesting a potential role for GA_4_ in bud break. Given our finding that floral initiation and reproductive shoot development occur concurrently in *C. oleifera* and previous reports stating that GA_4_ promotes dormancy release in the flower buds of Japanese apricot [52], we hypothesize that trace GA_4_ levels contribute to initiating floral and shoot development.

GA_3_ is known to regulate floral initiation and flowering time in plants, but the regulatory results are different or even opposite for different plants [53,54]. In our study, GA_3_ concentrations varied throughout floral development, with lower levels observed during the sprouting stage than during the dormancy and petal differentiation stages. This suggests that a low GA concentration may suffice for floral initiation in *C. oleifera*, whereas higher concentrations are required for petal differentiation. This also implied that GA_3_ may primarily influence petal development.

### 3.4. GA20OX1 and GA2OX8 Are the Key Candidate Genes in GA-Mediated Regulation of Floral Initiation by Gibberellin Pathway

*GA20OX1* and *GA3OX* are critical enzymes for the biosynthesis of bioactive GAs, which in turn regulate flowering and other developmental processes [55]. The balanced activity of genes associated with GA biosynthesis and GA inactivation, like *GA2OX*, is essential for maintaining optimal GA levels and proper developmental timing by allowing plants to respond appropriately to environmental cues and internal signals [55,56]. Excessive and insufficient GA levels can both disrupt the timing and efficiency of flowering [57,58]. Our results detected bioactive GAs (GA_3_ and GA_4_) and their biosynthetic precursors (GA_15_, GA_19_, GA_9_, and GA_20_) during floral initiation. Integrative analysis of endogenous GA monomer concentrations, DEGs involved in GA biosynthesis, and those involved in GA signal transduction pathways identified *GA20OX1* as a central gene regulating GA_3_ synthesis during floral initiation (Figure 8). Moreover, a positive correlation was found between GA_4_ levels and *GA2OX8* expression, suggesting that *GA2OX8* predominantly modulates GA_4_ levels. Notably, GA signaling-related genes exhibited elevated expression in early developmental stages, aligning with spring shoot sprouting—a critical phase of development. These findings suggest that low levels of active GAs are sufficient for spring shoot sprouting and floral initiation, whereas higher levels are required for floral organ development. This insight could inform targeted application of exogenous GAs to regulate distinct developmental phases in *C. oleifera*.

## 4. Materials and Methods

### 4.1. Plant Growth, Sample Collection, and Morphological Characteristics Analysis

The adult *Camellia oleifera* (cultivar ‘changlin53’) were grown under natural conditions in an experimental field at Experimental Center for Subtropical Forestry, Chinese Academy of Forestry (ECSF, CAF), which is located at Fenyi county, Jiangxi Province, China (27°49′ N, 114°39′ E; altitude 88–92 m). The different developmental stages of experimental materials were collected from February to June, and detailed information is shown in Table 1. After sampling, the tissues were quickly frozen in liquid nitrogen and stored at −80 °C until RNA isolation. Three biological replicates were used for each of the sampling points, and each biological replicate involved three mixed plant samples. Morphological characteristics analysis was performed according to Guo [59].

### 4.2. Quantitative Analysis of Endogenous Gibberellins

All biological specimens were cryogenically pulverized in liquid nitrogen. Precisely 50 mg of homogenized tissue was weighed (±0.1 mg accuracy) and supplemented with isotope-labeled internal standards. Metabolite extraction was carried out using 1 mL of methanol/water/formic acid (15:4:1, *v*/*v*/*v*) with vortex agitation for 30s, followed by ultrasonication (25 °C, 15 min). The supernatant was evaporated to dryness under a nitrogen stream (40 °C) and reconstituted in 100 μL of 80% methanol/water (*v*/*v*). The solution was filtered through a 0.22 μm Millipore filter prior to UPLC-MS/MS analysis. Quantitative profiling was conducted using ultra-performance liquid chromatography (UPLC; ExionLC™ AD system, SCIEX, https://sciex.com.cn/) coupled with tandem mass spectrometry (MS/MS; QTRAP^®^ 6500+ system, SCIEX, https://sciex.com.cn/). Ionization was performed via electrospray ionization (ESI) under the following conditions: ion source temperature of 550 °C, ion spray voltage of +5500 V (positive ion mode)/−4500 V (negative ion mode), and curtain gas (CUR) at 35 psi. The calibration curve’s construction was carried out as follows: A 12-point calibration series (0.01–500 ng/mL) was prepared by serial dilution of certified reference standards. Calibration curves were constructed by plotting the ratio of analyte-to-internal standard peak areas against their corresponding concentration ratios. Linear regression models (weighting factor: 1/x^2^) were validated, with R^2^ > 0.995 for all analytes. Endogenous hormone concentrations were calculated using the following formula:Concentration (ng/g) = c × v/1000/m
where C is the analyte concentration (ng/mL) derived from calibration curves based on integrated peak area ratios, V is the reconstitution volume (μL), and M is the sample’s mass (g).

### 4.3. Transcriptome Sequencing

Total RNA was extracted from all samples using the RNA prep Pure DP441 (Tiangen, Beijing, China), following the manufacturer’s protocol. The integrity and pollution were measured using agarose gel electrophoresis (AGE). The purity of the RNA samples was measured using a NanoDrop Micro-spectrophotometer (Thermo Scientifc, Waltham, MA, USA). The accurate concentration of the RNA samples was measured using a Qubit2.0 Fluorometer (Thermo Scientifc, Waltham, MA, USA). Accurate integrity was measured using an Agilent 2100 bioanalyzer (Agilent Technologies, Santa Clara, CA, USA). Clean data were obtained by removing reads containing adapters and removing poly-N and low-quality reads from raw data. Trinity software (Version v2.8.4) was used to assemble reads [60]. BUSCO software was used to evaluate the completeness of the assembly, and the results are shown in Appendix A [61].

### 4.4. Functional Annotation of the Transcriptome

Gene function was annotated using the following databases: Nr (NCBI non-redundant protein sequences), Swiss-Prot (a database of manually annotated and reviewed protein sequences), Pfam (Protein family), COG/KOG (Clusters of Orthologous Groups of proteins), KEGG (the Kyoto Encyclopedia of Genes and Genomes), and the GO (Gene Ontology) database.

### 4.5. Differentially Expressed Genes

The DESeq2 R package (v1.20.0, parameters set to default)was used to conduct pairwise comparisons. Read counts for each gene were normalized to reads per kilobase per million reads (RPKMs), which is currently the most widely used method for estimating gene expression levels and represents the gene expression levels [62]. There are strict algorithms for screening differentially expressed genes between two samples. Based on the RPKM values, the differential expression gene number across ten pairwise comparisons were calculated (Figure 3b,c), and multiple comparisons using false discovery rate (FDR) tests were conducted between two samples. A corrected FDR value of < 0.05 and |Fold change| > 2 was used as the criteria to judge the significant difference of gene expression.

### 4.6. GO and KEGG Enrichment Analysis of DEGs and Construction of Coexpression Network

We performed KEGG and GO enrichment analysis to explore the relevant pathway and biological functions that were activated during flower transition in Camellia oleifera. The background documents of GO and KEGG were from the GO and KEGG annotation. The GO and KEGG enrichment analysis of DEGs was performed using OmicShare tools, a free online platform for data analysis (http://www.omicshare.com/tools, 30 September 2024). A gene coexpression network was built using the WGCNA package. Parameters were set up as power = 12, minModuleSize = 50, and mergeCutHeight = 0.15. The networks were visualized using Cytoscape (Version 3.2.0).

### 4.7. Quantitative Real-Time PCR (qRT-PCR) Analysis

The total RNA was extracted from the same plant used for RNA sequencing. The cDNA was synthesized using a PrimeScriptTM RT reagent kit with gDNA Eraser (Perfect Real Time) (Takara Biomedical Technology, Beijing, China), following the manufacturer’s protocol. The cDNA reverse-transcription products were used as templates for qRT-PCR. qRT-PCR was performed using TB Green^®^ Premix Ex Taq™ II (Tli RNaseH Plus) (TaKaRa), according to the manufacturer’s instructions. The calculation of the relative expression of the target gene in each sample was conducted using the 2−ΔΔCt method. The relative expression values of qRT-PCR and RPKM values of RNA-seq were compared. All primers for qRT-PCR are listed in Appendix A.

## 5. Conclusions

This study presents a comprehensive comparison of morphological traits, endogenous GA concentrations, and transcriptomic profiles of buds at various developmental stages during floral transition in adult *C. oleifera* ‘changlin53’. Six types of GAs were detected, among which GA_4_ and GA_3_ appeared to play distinct roles: GA_4_ was associated with floral initiation and bud break, while GA_3_ was linked to petal differentiation. Reproductive shoot development is a crucial aspect of reproductive growth initiation in the annual cycle of *C. oleifera*, with floral initiation and reproductive shoot growth occurring simultaneously and at the same location. The photoperiod pathway involving *CoCO-like*, *CoFT*, and *CoLFY* was identified as a key regulator of floral initiation, activating the transition of the stem apical meristem into a floral meristem. Combined morphological and transcriptomic analyses indicate that floral initiation in *C. oleifera* is a multigenic and continuous regulatory process. Among these, the *FT-LFY* regulatory module plays a central role in orchestrating the transition from vegetative to floral development.

## Figures and Tables

**Figure 1 plants-14-02348-f001:**
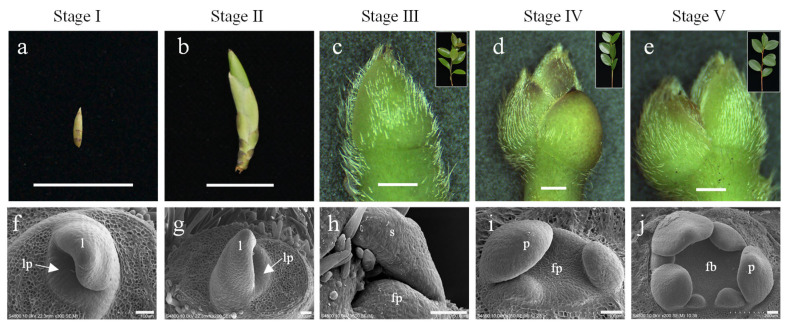
Developmental stages of buds in *C. oleifera*. Buds (**a**,**b**) are referred to the vegetative bud formed in the last year. Buds in stages I are distinguished as follows: (**a**) developed into sprouted buds (**b**); only vegetative buds grown (**f**,**g**); no flower buds occurred in stages I and II. Buds (**c**–**e**) in stages III, IV, and V were the apical buds of spring shoots (panel **c**–**e**), which were developed from a vegetative bud (**b**). Flower buds (**h**–**j**) in III, IV, and V were located at the base of apical buds (**c**–**e**). lp: Leaf primordium; fp: floral primordium; l: leaf, s: sepal; p: petal. The scale bars in a to b, in c to e, and in f to j indicate 1.5 cm, 1000 μm, and 50 μm, respectively.

**Figure 2 plants-14-02348-f002:**
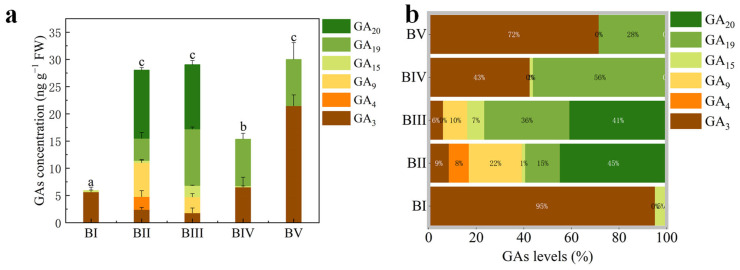
Concentrations of endogenous gibberellin types (**a**) and their relative proportions (**b**) at different developmental stages. The letters on the top of column indicate the multiple comparison results of total GAs concentration at 0.05 level.

**Figure 3 plants-14-02348-f003:**
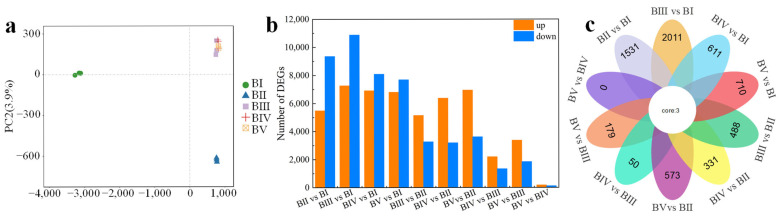
Principal component analysis of the RNA-seq data in all samples (**a**). Numbers of DEGs in diverse pairwise comparisons (**b**) and Venn diagram showing the common and unique DEGs among different pairwise comparisons (**c**).

**Figure 4 plants-14-02348-f004:**
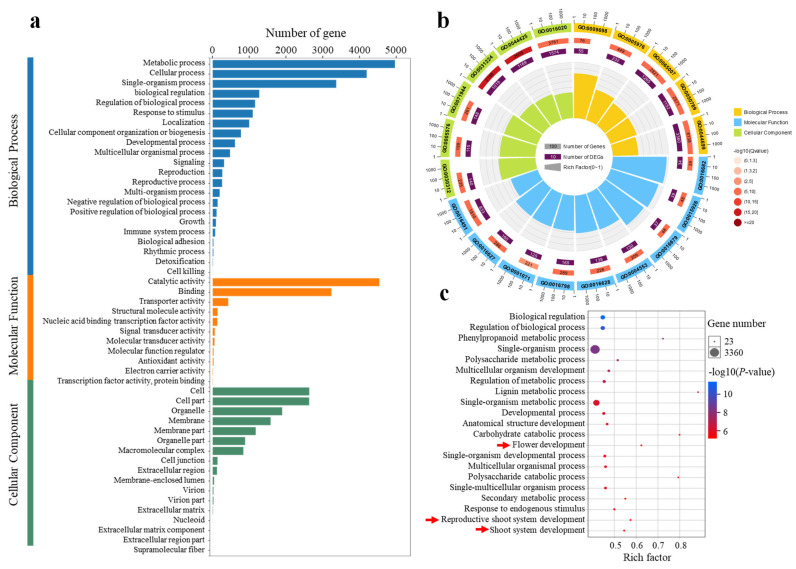
GO functional of the significantly differentially expressed genes (DEGs) in diverse pairwise comparisons (**a**). Top 20 significance-level GO terms of DEGs (**b**). Top 21 significance-level GO terms of BP (**c**). The red arrow indicates the GO term related to floral initiation.

**Figure 5 plants-14-02348-f005:**
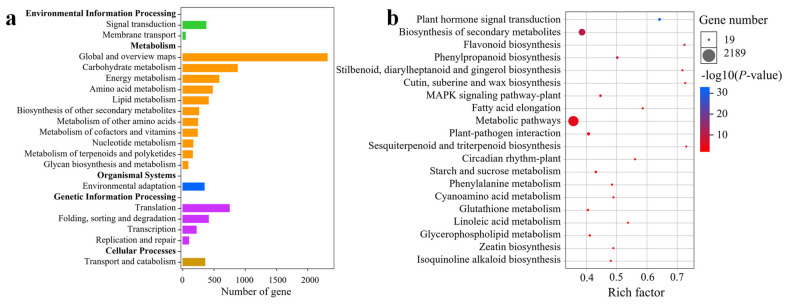
KEGG pathway enrichment analysis of the significantly DEGs of buds in diverse pairwise comparisons (**a**). Top 20 significance-level KEGG pathways of DEGs in buds (**b**).

**Figure 6 plants-14-02348-f006:**
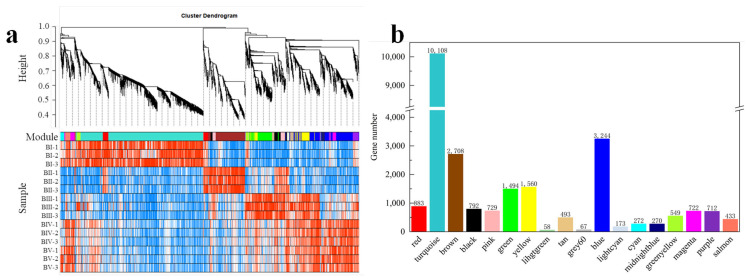
Hierarchical cluster tree showing co-expression modules identified by WGCNA (**a**) and the DEG number of each module (**b**).

**Figure 7 plants-14-02348-f007:**
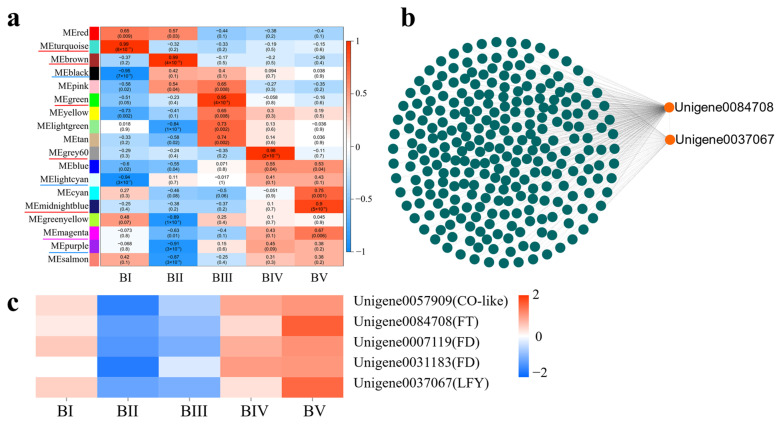
Relationship between modules and developmental stages (**a**). Each row represents a module, and each column represents a developmental stage. The intersection cells display correlation coefficients and *p*-values between the modules and stages. Red cells indicate strong positive correlations between a specific module and a developmental stage, whereas blue cells indicate strong negative correlations. Modules with significant positive or negative correlations are underlined in red or blue, respectively. Magenta underlines denote modules with distinct correlation patterns. Co-expression network analysis of DEGs in the magenta module (**b**). Expression heatmap of candidate hub genes related to floral initiation in the magenta module (**c**).

**Figure 8 plants-14-02348-f008:**
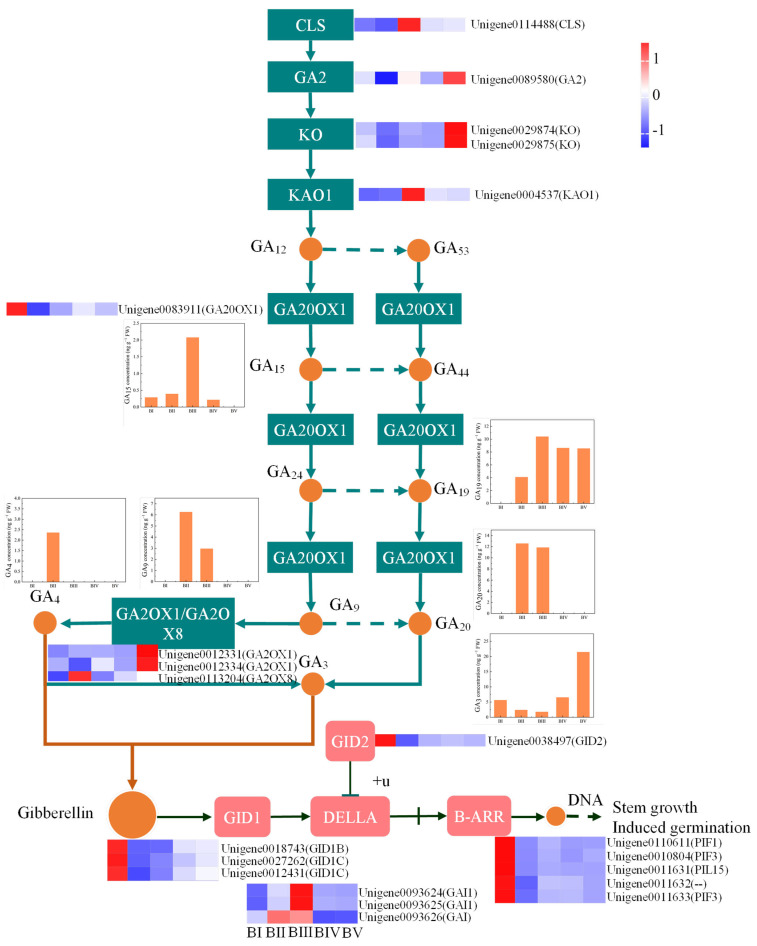
Comprehensive analysis of DEGs and endogenous GA monomer concentrations in GA biosynthesis and GA signal transduction pathways. The box plot shows the concentration of different gibberellin monomers at different developmental stages in the GA biosynthesis pathway. The heatmap demonstrated the expression levels of DEGs in the GA synthesis pathway and the signal transduction pathway during different developmental stages.

**Table 1 plants-14-02348-t001:** Overview of the samples for RNA-seq.

Developmental Stages	Sample Name	Number of Clean Reads	Number of Clean Data	Q20 (%)	Q30 (%)	GC (%)	Total Mapped (%)
I	BI-1	56,907,348	8,497,755,633	8,338,100,557 (98.12%)	8,015,960,394 (94.33%)	3,900,291,117 (45.90%)	44,985,690 (79.05%)
	BI-2	48,431,060	7,215,452,943	7,085,854,082 (98.20%)	6,822,179,906 (94.55%)	3,339,480,736 (46.28%)	38,526,147 (79.55%)
	BI-3	58,652,190	8,740,998,410	8,563,905,357 (97.97%)	8,212,926,590 (93.96%)	4,031,458,880 (46.12%)	45,998,276 (78.43%)
II	BII-1	54,553,160	8,112,660,091	7,889,813,970 (97.25%)	7,477,825,967 (92.17%)	3,768,480,169 (46.45%)	44,160,960 (80.95%)
	BII-2	44,576,422	6,624,937,074	6,457,818,501 (97.48%)	6,135,698,351 (92.62%)	3,151,122,729 (47.56%)	37,181,979 (83.41%)
	BII-3	46,778,886	6,962,446,522	6,759,631,130 (97.09%)	6,392,012,670 (91.81%)	3,234,932,717 (46.46%)	37,839,837 (80.89%)
III	BIII-1	50,985,660	7,586,849,110	7,365,728,863 (97.09%)	6,964,564,010 (91.80%)	3,511,913,268 (46.29%)	41,408,399 (81.22%)
	BIII-2	50,562,454	7,545,683,270	7,317,183,863 (96.97%)	6,915,262,966 (91.65%)	3,461,771,628 (45.88%)	40,696,612 (80.49%)
	BIII-3	46,393,600	6,891,860,986	6,708,819,561 (97.34%)	6367327747 (92.39%)	3,208,481,881 (46.55%)	38,154,413 (82.24%)
IV	BIV-1	41,886,036	6,228,273,002	6,066,396,478 (97.40%)	5,763,909,687 (92.54%)	28,57,941,556 (45.89%)	33,925,334 (80.99%)
	BIV-2	51,563,192	7,699,619,603	7485225086 (97.22%)	7,085,303,270 (92.02%)	3,537,890,895 (45.95%)	42,049,101 (81.55%)
	BIV-3	52,264,300	7,811,770,783	7614802319 (97.48%)	7,238,347,545 (92.66%)	3,592,384,741 (45.99%)	42,578,151 (81.47%)
V	BV-1	49,788,996	7,439,686,499	7,227,532,557 (97.15%)	6,835,950,341 (91.88%)	3,417,730,361 (45.94%)	40,375,415 (81.09%)
	BV-2	48,638,636	7,269,786,502	7,078,040,967 (97.36%)	6,716,162,908 (92.38%)	3,349,964,297 (46.08%)	39,285,275 (80.77%)
	BV-3	57,903,406	8,647,231,084	8,394,242,589 (97.07%)	7,932,070,431 (91.73%)	3,978,878,410 (46.01%)	46,499,461 (80.31%)

## Data Availability

Data are contained within the article and Appendix A.

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
