# Peer review of "Transcriptomic Profiling of Buds Unveils Insights into Floral Initiation in Tea-Oil Tree (Camellia oleifera ‘changlin53’)"

_plants, 2025, doi:10.3390/plants14152348_

Round 1
Reviewer 1 Report
Comments and Suggestions for Authors
In this paper, the research background is fully elaborated, the research design is appropriate, the method is fully described, the results are clearly presented, the results can support the conclusions, and the figures and tables are clear and well presented.
1.Figure 1. The description of vegetative bud and flower bud differentiation stages is not clear and difficult to understand.
2.Figure 1. ‘ lp: leaf primordium, fp: floral primordium’,however, Ip and FP are not shown in Figure 1.
Author Response
Reviewer 1:
In this paper, the research background is fully elaborated, the research design is appropriate, the method is fully described, the results are clearly presented, the results can support the conclusions, and the figures and tables are clear and well presented.
General answer: Thank you for reviewing our manuscript. Your positive comments and suggestions are highly appreciated.
Specific comments:
Comments 1: Figure 1. The description of vegetative bud and flower bud differentiation stages is not clear and difficult to understand.
Response 1: Thanks for your professional comments. In the early stage of the floral initiation in Camellia oleifera, it is very difficult to distinguish between the vegetative bud and flower bud differentiation stages. Before this, I wrote a separate article to clearly define this distinction (DOI:10.13275/j.cnki.lykxyj.2022.03.014).
Comments 2: Figure 1. ‘ lp: leaf primordium, fp: floral primordium’,however, Ip and FP are not shown in Figure 1.
Response 2: Thanks for your careful reading. I'm very sorry that there were two writing errors in Figure 1. We have respectively changed "lb" and "fb" to "lp" and "fp" in Figure 1.
Reference for this answer:
Hong-yan, G.; Xin-jian, T.; Feng, T.; Qiu-ping, Z., The Relationship between Floral Initiation and Spring-shoot Growth in Camellia oleifera. Forest Research 2022, 35, 123-130. (DOI:10.13275/j.cnki.lykxyj.2022.03.0 14)
Reviewer 2 Report
Comments and Suggestions for Authors
Review of ms. plants-3753963 entitled Transcriptomic Profiling of Buds Unveils Insights into Floral Initiation in Tea-Oil tree (Camellia oleifera ‘changlin53’)
This manuscript reports on the analysis of morphology and transcriptomic experiments of floral initiation in Camellia oleifera ‘changlin53’. By integrating UPLC-MS/MS-based hytohormone profiling with high-quality RNA-seq data and applying robust bioinformatic approaches such as WGCNA, the study provides valuable insights into the molecular regulation of floral transition in this economically important oil-producing tree.
The manuscript is clearly structured, the methodology is adequately described, and the conclusions are well-supported by the results. The identification of components of the FT-LFY regulatory module and gibberellin-associated candidate genes represents a meaningful contribution to the understanding of flowering regulation in woody perennials.
I still see that there are shortcomings in it that could be corrected with minor improvements:
- While the Introduction provides a comprehensive overview of the current knowledge on floral initiation in both model plants and woody perennials, it lacks a clearly stated hypothesis or research question. The authors outline the importance of floral initiation in Camellia oleifera and summarize known regulatory pathways (e.g., photoperiod, gibberellin, FT-LFY modules), but they do not explicitly articulate what specific mechanisms or gene networks they aim to test or validate in their study.
- For clarity and scientific rigor, the authors should explicitly formulate a hypothesis—e.g., that floral initiation in C. oleifera is regulated by conserved flowering pathways involving FT, LFY, and GA biosynthesis/signaling genes, similar to those known in model species. Additionally, it would be helpful if they briefly described how their methodological approach (e.g., morphological staging, UPLC-MS/MS GA quantification, RNA-seq, WGCNA) is designed to test this hypothesis in the context of existing literature.
- I suggest to complete the Introduction with more specific literature information on the evidence of the phytohormone signaling and morphogenesis, floral development is tightly regulated by dynamic, stage-specific hormonal signaling networks and transcriptional programs (e.g. Virág et al. generated a high-resolution temporal transcriptome atlas across defined developmental stages of soybean inflorescences, identifying waves of gene expression—particularly in hormone-related pathways such as auxin, gibberellin, cytokinin, ABA, ethylene, and jasmonate. (https://doi.org/10.3390/ijms26136455).
- The manuscript does not address the floral sexual morphology or the potential presence of sex-specific differentiation during floral initiation and development. It is important to note that Camellia oleifera is a monoecious species with hermaphroditic (bisexual) flowers, as documented in botanical literature. Therefore, the flowers contain both stamens and pistils. Despite this, the manuscript does not mention or analyze the sexual organ development, nor does it examine whether any sex-specific gene expression patterns are involved in the floral initiation process. Given that floral development inherently includes the establishment of sexual organs, and considering the transcriptomic depth of this study, the authors are encouraged to briefly address: whether any genes related to male or female organ identity (e.g., B- or C-class MADS-box genes) were identified; whether sexual differentiation is detectable at any of the analyzed developmental stages; and whether this process appears synchronous or coordinated within the flower primordia.
Even if C. oleifera flowers are bisexual and do not require sex-specific regulation per se, a short discussion or confirmation of this biological context would enhance the completeness and interpretative clarity of the study.
- Ensure that figure legends are self-contained. For instance, Figure 7’s legend should clearly state what the color codes (red/blue/magenta underlines) represent.
- Figure 8 should be presented in a more readable form.
- BUSCO statistics on transcriptome completeness are mentioned but not shown—please include this in supplementary data or the main text.
- In section 2.5, please explain whether the identified common DEGs were validated further (e.g., via qRT-PCR or co-expression analysis).
- In conclusion, I suggest explaining practical applications of the results (e.g., hormone treatment timing) based on the GA results.
Author Response
Comments 1: While the Introduction provides a comprehensive overview of the current knowledge on floral initiation in both model plants and woody perennials, it lacks a clearly stated hypothesis or research question. The authors outline the importance of floral initiation in Camellia oleifera and summarize known regulatory pathways (e.g., photoperiod, gibberellin, FT-LFY modules), but they do not explicitly articulate what specific mechanisms or gene networks they aim to test or validate in their study.
Response 1: Thanks for your professional comments. In the introduction section of the manuscript, we have added the hypothesis about floral initiation network in Camellia oleifera. “We revealed that old leaves in C. oleifera serve as key photoperiodic sensitive organs, promoting FT expression through light-responsive mechanisms to initiate floral bud formation[1]. The transduction of endogenous signaling molecules from old leaves to axillary buds is required to initiate floral transition through induced transcriptional activation of floral meristem-specific genes. However, the specific genetic determinants mediating this developmental shift remain to be systematically characterized through comprehensive transcriptomic analysis and which genes in the buds were activated? LFY, AP1 or any other genes . ”(the last paragraph of introduction)
Reference for this answer:
[1] Guo, H.; Zhong, Q.; Tian, F.; Zhou, X.; Tan, X.; Luo, Z. Transcriptome Analysis Reveals Putative Induction of Floral Initiation by Old Leaves in Tea-Oil Tree (Camellia oleifera ‘changlin53’). Int. J. Mol. Sci. 2022, 23, 13021. https://doi.org/10.3390/ijms232113021
Comments 2: For clarity and scientific rigor, the authors should explicitly formulate a hypothesis—e.g., that floral initiation in C. oleifera is regulated by conserved flowering pathways involving FT, LFY, and GA biosynthesis/signaling genes, similar to those known in model species. Additionally, it would be helpful if they briefly described how their methodological approach (e.g., morphological staging, UPLC-MS/MS GA quantification, RNA-seq, WGCNA) is designed to test this hypothesis in the context of existing literature.141-142. In this part, the author describes transcriptome sequencing with the De novo assembly of these clean reads. At present, the genome information of Camellia oleifera is very complete. Why didn't the author refer to known transcription?
Response 2: Thanks for your constructive comments. This study sampled in 2019 and performed transcriptome analysis in 2020. At that time, there was no reference genome, so no reference transcriptome analysis was performed. After publication of the Camellia oleifera genome and Camellia. Lanceoleosa genome in 2022, we performed our results with their homology and referred to their annotation information, and found that the annotation results were similar, especially in significantly DEGs. So, we adopted the initial analysis results.
Comments 3: I suggest to complete the Introduction with more specific literature information on the evidence of the phytohormone signaling and morphogenesis, floral development is tightly regulated by dynamic, stage-specific shormonal signaling networks and transcriptional programs (e.g. Virág et al. generated a high-resolution temporal transcriptome atlas across defined developmental stages of soybean inflorescences, identifying waves of gene expression—particularly in hormone-related pathways such as auxin, gibberellin, cytokinin, ABA, ethylene, and jasmonate. (https://doi.org/10.3390/ijm s26136455).
Response 3: Thanks for your constructive comments. Floral initiation is influenced by both external and internal signals. Environmental factors, such as photoperiod, temperature, nutrient status, drought, salinity, exogenous hormones and chemicals, and microbial pathogens, as well as endogenous signals, such as plant age, hormone levels, and carbohydrate status, all contribute to its regulation. In this manuscript, we mainly discussed the transcriptional regulation within the buds and the gibberellin-related pathways. In addition, we also conducted research on the relationship between hormones such as auxin, gibberellin, cytokinin, ABA, ethylene, and jasmonate and floral initiation. There are complex interrelationships among these hormones with floral initiation, and this will be elaborated on in a new manuscript.
Comments 4: The manuscript does not address the floral sexual morphology or the potential presence of sex-specific differentiation during floral initiation and development. It is important to note that Camellia oleifera is a monoecious species with hermaphroditic (bisexual) flowers, as documented in botanical literature. Therefore, the flowers contain both stamens and pistils. Despite this, the manuscript does not mention or analyze the sexual organ development, nor does it examine whether any sex-specific gene expression patterns are involved in the floral initiation process. Given that floral development inherently includes the establishment of sexual organs, and considering the transcriptomic depth of this study, the authors are encouraged to briefly address: whether any genes related to male or female organ identity (e.g., B- or C-class MADS-box genes) were identified; whether sexual differentiation is detectable at any of the analyzed developmental stages; and whether this process appears synchronous or coordinated within the flower primordia.
Response 4: Thanks for your constructive comments. It takes more than 8 months from bud differentiation to blooming in Camellia oleifera .We conducted research in three stages: floral initiation, floral organ development, and flowering. In this manuscript, we mainly discussed the transcriptional regulation and the gibberellin-related pathways at floral initiation stage. In the future research plan, we will focus on the floral organ development firstly, then study the induction of flower blooming. Finally, we will jointly analyze the regulatory mechanisms of these three parts including floral initiation, floral organ development, and flowering.
Comments 5: Ensure that figure legends are self-contained. For instance, Figure 7’s legend should clearly state what the color codes (red/blue/magenta underlines) represent.
Response 5: Thanks for your careful reading. In the caption of Figure 7, we explained the meanings represented by the different colored lines.”Each row represents a module, and each column represents a developmental stage. The intersection cells display correlation coefficients and P-values between the modules and stages. Red cells indicate strong positive correlations between a specific module and a developmental stage, whereas blue cells indicate strong negative correlations. Modules with significant positive or negative correlations are underlined in red or blue, respectively. Magenta underlines denote modules with distinct correlation patterns.”
Comments 6: Figure 8 should be presented in a more readable form
Response 6: Thanks for your professional comments. Figure 8 shows the concentration of gibberellin monomers in buds at different developmental stages during gibberellin synthesis, the expression levels of DEGs, and the expression levels of DEGs in the gibberellin signal transduction pathway. To make this figure more understandable for readers, we have provided detailed explanations in the annotations.”The box plot shows the concentration of different gibberellin monomers at different developmental stages in GA biosynthesis pathway. The heatmap demonstrated the expression levels of DEGs in the GA synthesis pathway and the signal transduction pathway during different developmental stages.”
Comments 7: BUSCO statistics on transcriptome completeness are mentioned but not shown—please include this in supplementary data or the main text.
Response 7: Thanks for your professional comments. We showed the BUSCO statistics in supplementary data.
Comments 8: In section 2.5, please explain whether the identified common DEGs were validated further (e.g., via qRT-PCR or co-expression analysis).
Response 8: Thanks for your professional comments. In previous studies, we identified through phenotypic characteristics that the spring shoots and flower initiation occurred simultaneously. In this manuscript, we also obtained the same results through transcriptome analysis, thereby providing another perspective to further confirm the previous viewpoint (DOI:10.13275/j.cnki.lykxyj.2022.03.014).
Reference for this answer:
Hong-yan, G.; Xin-jian, T.; Feng, T.; Qiu-ping, Z., The Relationship between Floral Initiation and Spring-shoot Growth in Camellia oleifera. Forest Research 2022, 35, 123-130. (DOI:10.13275/j.cnki.lykxyj.2022.03.0 14)
Comments 9: In conclusion, I suggest explaining practical applications of the results (e.g., hormone treatment timing) based on the GA results.
Response 9: Thanks for your professional comments. We have applied for a new project to verify the results of this part. Please waiting for our team's manuscripts in the next three years.
Reviewer 3 Report
Comments and Suggestions for Authors
This manuscript focused on revealing the genes related to the floral initiation of flower buds in Camellia oleifera through transcriptome analyses. The flower bud transcriptome data and GA analysis, as determined by the authors at various time points, provide evidence to substantiate their arguments. I recommend that minor revisions be made to this manuscript. Please see the attached PDF.
1) In Abstract section, the developmental stages set by the authors are information known only to the authors. Add information about this: sprouting stage (B II)
2) vernalization
3) Please ensure that there is spacing between words and reference numbers.
4) Please be mindful of the appropriate use of italics: FT is repressed by~
5) The second time an acronym appears, write it as an abbreviation: LEAFY -> LFY
6) In Figure 1 caption, only the V seems to have a different font.
7) Add (Figure S1) right here.
8) In Figure 5 caption, where is (a)?
9) DEGs?
10) Write in italics.
11) DESeq2 R package

Author Response
Comments 1: In Abstract section, the developmental stages set by the authors are information known only to the authors. Add information about this: sprouting stage (B II).
Response 1: Thanks for your professional comments. I have revised the expression here and also made changes to another part in the abstract.
Comments 2: vernalization
Response 2: Thanks for your careful reading. I have corrected the spelling mistake.
Comments 3: Please ensure that there is spacing between words and reference numbers.
Response 3: Thanks for your careful reading. The modifications have been made in accordance with the formatting requirements of the journal.
Comments 4: Please be mindful of the appropriate use of italics: FT is repressed by~
Response 4: Thanks for your careful reading. The modifications have been completed.
Comments 5: The second time an acronym appears, write it as an abbreviation: LEAFY -> LFY
Response 5: Thanks for your careful reading. The modifications have been completed.
Comments 6: In Figure 1 caption, only the V seems to have a different font
Response 6: Thanks for your careful reading. The modifications have been completed.
Comments 7: Add (Figure S1) right here
Response 7: Thanks for your careful reading.The modifications have been completed.
Comments 8: In Figure 5 caption, where is (a)?
Response 8: Thanks for your careful reading. In Figure 5 , It (a) was added to the appropriate position.
Comments 9: DEGs?
Response 9: Thanks for your careful reading. It has been confirmed there is "DEGs".
Comments 10: Write in italics
Response 10: Thanks for your careful reading. The modifications have been completed.
Comments 11: DESeq2 R package
Response 11: Thanks for your careful reading. The modifications have been completed.